# Different Vitamin D Supplementation Strategies in the First Years of Life: A Systematic Review

**DOI:** 10.3390/healthcare10061023

**Published:** 2022-06-01

**Authors:** Antonio Corsello, Gregorio Paolo Milani, Maria Lorella Giannì, Valeria Dipasquale, Claudio Romano, Carlo Agostoni

**Affiliations:** 1Department of Clinical Sciences and Community Health, University of Milan, 20122 Milan, Italy; antonio.corsello@unimi.it (A.C.); maria.gianni@unimi.it (M.L.G.); carlo.agostoni@unimi.it (C.A.); 2Pediatric Unit, Fondazione IRCCS Ca’ Granda Ospedale Maggiore Policlinico, 20122 Milan, Italy; 3Neonatal Intensive Care Unit, Fondazione IRCCS Ca’ Granda Ospedale Maggiore Policlinico, 20122 Milan, Italy; 4Pediatric Gastroenterology and Cystic Fibrosis Unit, Department of Human Pathology in Adulthood and Childhood “G. Barresi”, University of Messina, 98125 Messina, Italy; dipasquale.valeria@libero.it (V.D.); claudio.romano@unime.it (C.R.)

**Keywords:** Vitamin D, Vitamin D supplementation, Vitamin D insufficiency, infant, micronutrients, toddlers, breast milk, breastfeeding, formula feeding, daily supplementation adherence

## Abstract

Vitamin D (VD) is an essential micronutrient with multiple functions for human growth, and adequate intake should be guaranteed throughout life. However, VD insufficiency is observed in infants all over the world. Low VD concentration in the breast milk of non-supplemented mothers and low compliance to VD daily supplementation are the main causes of VD insufficiency, especially in the long term. Furthermore, VD supplementation dosages are still debated and differ by country. We conducted a systematic review to compare the most recent evidence on different postnatal VD supplementation strategies, determining whether supplementation given to the mother is as effective as that administered directly to the child, and whether different dosages and administration schedules differ significantly in terms of efficacy and safety. We identified 18 randomized controlled trials (RCTs) addressing the role of infant (n = 961), maternal (n = 652) or combined infant and maternal VD supplementation (n = 260 pairs). In all studies, similar outcomes emerged in terms of efficacy and safety. According to our findings, alternative approaches of VD supplementation may be adopted, especially in cases where the adherence to daily supplementation strategies is poor. This review shows that different dosages and supplementation strategies result in similar VD sufficiency rates. Therefore, international guidelines may be revised in the future to offer multiple and different options of supplementation for specific settings and ages.

## 1. Introduction

Human breast milk, despite being the best choice for infant feeding, is generally lacking in proper amounts of Vitamin D (VD), with concentrations ranging from 10 to 80 IU/L in healthy lactating women [1]. Since the amount of breast milk VD is correlated with circulating levels of the vitamin, mothers with a poor VD status supply their infants with even less of these amounts. On the other hand, high serum levels of 25-hydroxyvitamin D (25(OH)D) in mothers correlate with higher breast milk amounts of VD [2,3,4]. The relevance of VD has changed in recent years, thanks to the knowledge of the many health outcomes VD has been associated with, especially regarding early development, bone health and immune system [5]. Therefore, by virtue of the VD functional role and the possible negative impact of its deficiency, it is important to ensure adequate levels in newborns, breastfed infants and toddlers, when a proper growth and nutrition is essential, and needs may change quickly [6]. For these reasons, the American Academy of Pediatrics and many European societies advocate VD supplementation to mothers, their children or both of them in order to prevent the risk of VD deficiency, which may be increased due to the limited sun exposure typical of the first months of life [7].

Furthermore, it is known that VD serum levels are maintained by cutaneous synthesis and oral intake, while VD metabolism is both renal and hepatic [8]. Nonetheless, multiple factors affect the overall VD status, with seasonality, clouds, altitude and latitude all having a direct impact on the sun angle and ultraviolet radiations [9]. Other factors that may significantly alter the VD status include genetic predispositions, skin type (skins with high concentrations of melanin need more time of sunlight exposure to achieve the same levels of VD sufficiency) and obesity [10]. In fact, both food absorption and cutaneous synthesis have been found to be linked to different quantities of adipose tissue [11]. This evidence could be related to the altered bioavailability of liposoluble vitamins in subjects with higher fat mass or high hepatic lipid content, which has been shown to be inversely correlated with VD serum concentrations [12].

A wide variety of methods and dosages of supplementation are described in the literature. Oral supplementation of 400 IU/day to all infants from few days after birth to at least 6 months of age is generally recommended [13]. However, a high prevalence of VD insufficiency is observed worldwide among infants, possibly due to caregivers’ poor compliance and acceptance of daily supplementation, with concurrent formulation heterogeneity [14,15,16]. Higher maternal postpartum supplementation or intermittent infant oral intake at higher doses could then represent a valid alternative [17,18]. However, the best supplementation strategy in the first period of life is still debated [19,20,21].

Furthermore, a low adherence to constant daily supplementation in infants and toddlers, even in culturally and economically diverse settings, has been frequently described [22,23,24,25]. For this reason, in addition to the lack of evidence about a substantial difference between a daily or a weekly/monthly administration or possible adverse effects, we decided to analyze different VD supplementation methods, comparing the VD sufficiency rates that followed the different interventions, while considering both safety and efficacy outcomes.

## 2. Materials and Methods

### 2.1. Search Strategy

This systematic review of the literature was conducted according to the Preferred Reporting Items for Systematic reviews and Meta-Analyses (PRISMA) guidelines [26]. Three different computerized databases (PubMed, the Cochrane Library and Web of Science) were searched using the following string: (children OR infants OR toddlers OR pediatric) AND (vitamin D) AND (trial OR supplementation) AND (breastfeeding OR feeding OR complementary OR milk OR formula OR lactation). The literature search was conducted on 1 October 2021.

### 2.2. Study Selection

Studies were included if they met the inclusion criteria of being conducted on healthy children aged from 0 to 36 months, born full-term to mothers on a VD sufficiency status at the time of delivery and coming from a developed country. Only randomized controlled clinical trials (RCTs) published in English between 1 January 2000 and 30 September 2021 were considered.

Exclusion criteria were: (i) observational studies; (ii) RCTs with fewer than 15 mother–infant pairs; (iii) children older than 36 months; (iv) cohorts coming from less developed countries; (v) prenatal VD supplementation; (vi) baseline maternal 25(OH)D concentrations less than 10 ng/mL (severe deficiency); and (vii) absence of evaluation of VD status in children as a main outcome of the study.

The titles and abstracts of articles that met the inclusion criteria were then independently screened for relevance by two authors, and the full text of potentially relevant articles was evaluated in depth by the same two investigators. Investigators were chosen for their expertise in the field, particularly in pediatric nutrition and supplementation policies.

### 2.3. Analyses and Endpoints

A quality assessment of the studies was conducted in order to assess their most important biases and weaknesses. The Newcastle-Ottawa Scale was used to assess the quality of the observational studies [27]. This scale evaluates the studies according to the comparability of the results, the selection of the population and the controls, and the reliability of the outcomes. Eventual discrepancies in the quality assessment were discussed and resolved by three independent authors. Parameters considered for a high-quality evaluation were the selection of the population, the comparability of the study and its outcomes. Only studies with a high-quality evaluation were included in this review.

From each selected study, the following data were extracted using a predefined database: name of the first author, year of publication, country, number of participants, route of administration, dosage and timing of VD supplementation, length of follow-up and risk ratios.

The studies were then independently considered according to the following supplementation strategies: VD supplementation trials for children, for lactating mothers and for both lactating mothers and their children. Data on the accuracy and reliability of the assay for VD assessment were also extracted [28,29].

The primary study endpoint was to compare different methods and timing of VD supplementation on 25(OH)D serum concentrations after the intervention. The secondary endpoint was to assess any potential side effect related to the specific intervention among the different cohorts, when reported.

## 3. Results

For the final analysis, 18 articles were selected. The literature search process is summarized in Figure 1. The studies included were conducted in the following continents: seven in America, four in Asia, four in Europe and three in Oceania.

### 3.1. Child Supplementation

Eight trials investigated the supplementation of VD in children only [30,31,32,33,34,35,36,37]. The studies included a total of 961 subjects aging from 0 to 36 months (Table 1). Six of the eight studies included infants only (0–12 months of age), while the other two focused on toddlers (12–36 months of age).

Various supplementation regimens were assessed in these studies. Two studies evaluated the efficacy of a single high dose of 50,000 IU of VD given to infants at birth [34,35]. In one study, the control group received placebo for six months [34], while in the other study, controls received 400 IU VD daily for four months [35]. A single high dose of 50,000 IU resulted to be as much effective as daily VD administration of 400 IU, achieving average VD serum levels >20 ng/mL and resulting significantly effective compared with placebo [34,35]. Three further studies investigated the efficacy of the generally recommended daily dose of 400 IU as compared with placebo or inferior or higher daily doses of VD [30,31,32,33]. In the study comparing VD supplementation to placebo, the intervention group had significantly higher levels of VD than the placebo group (38.4 vs. 20.2 ng/mL at 3 months and 38 vs. 28.8 ng/mL at 6 months in breastfed infants). In the three studies comparing different VD dosages, no significant differences were found in the VD sufficiency ratio in infants, even if higher VD serum levels were reached in higher-intake groups [31,32,33]. A dose of 200 IU/d was shown to be insufficient to achieve adequate VD levels [33].

The studies conducted on toddlers included two cohorts comparing the administration of VD-fortified milk formula with cow’s milk or meat-derived food [36,37]. Both studies found that VD-fortified milk, especially in the second year of life, was significantly associated with an improvement in VD status. Average levels of 25(OH)D > 30 ng/mL were observed in both intervention groups. On the contrary, a higher rate of VD deficiency was found in toddlers receiving meat-derived food or cow’s milk when compared with children receiving VD-fortified milk (43% vs. 15% in the meat group and 33.7% vs. 13.5% in the cow’s milk group).
healthcare-10-01023-t001_Table 1Table 1Studies investigating Vitamin D supplementation for children.References (Authors, Year, Country)Patients, nIntervention (VD Prophylaxis) Timing of the Study (Since Delivery)Main OutcomeAdverse EventsAlonso et al. 2011, Spain [30]88 (n = 41 intervention group, n = 47 placebo group)400 IU/d; oralFrom 1 to 12 months25(OH)D levels significantly >35 ng/mL at 3 and 6 months in the intervention group onlyNoneHolmlund-Suila et al. 2012, Finland [31]93 (n = 29 group 1, n = 32 group 2, n = 32 group 3)400 IU/d (group 1), 1200 IU/d (group 2), 1600 IU/d (group 3); oralFrom 2 weeks to 3 monthsFinal serum 25(OH)D concentration >20 ng/mL in all children (regardless of VD dosing)NoneGallo et al. 2013, Canada [32]132 (n = 39 group 1, n = 39 group 2, n = 38 group 3, n = 16 group 4)400 IU/d (group 1), 800 IU/d (group 2), 1200 IU/d (group 3), 1600 IU/d (group 4); oralFrom 1 to 12 monthsAll dosages established 25(OH)D concentrations of >20 ng/mL or above in 97% of infants at 3 months and maintained this in 98% of infants for 12 monthsNoneZiegler et al. 2014, USA [33]142 (n = 43 group 1, n = 37 group 2, n = 36 group 3, n = 26 group 4)200 IU/d (group 1), 400 IU/d (group 2), 600 IU/d (group 3), 800 IU/d (group 4); oralFrom 1 to 10 monthsMost infants had low (<25 ng/mL) 25(OH)D levels at 1 month, although levels increased with supplementation; the highest doses were more efficacious in maintaining VD sufficiency, although the findings support the recommended 400 IU/d dose.NoneMoodley et al. 2015, Mexico [34]49 (n = 27 intervention group, n = 22 placebo group)Single dose of 50,000 IU; oralFrom 24 h after birth to 6 monthsSustained levels of VD >32 ng/mL through the first 6 months in the intervention groupNoneHuynh et al. 2017, Australia [35]70 (n = 36 group 1, n = 34 group 2)400 IU/d (group 1) vs. single 50,000 IU dose (group 2); oralFrom 0 to 4 monthsHigher 25(OH)D sufficiency rates (>20 ng/mL) at 1–2 weeks in group 2 compared with group 1; similar rates by 3–4 months NoneHoughton et al. 2011, New Zealand [36]181 (n = 107 intervention group, n = 74 control—meat derived food)VD fortified milk (40 IU/100 mL)From 12 to 20 monthsSignificantly lower prevalence of 25(OH)D level <20 ng/mL (insufficiency) in the intervention groupNot availableAkkermans et al. 2017, Germany/ Netherlands/ England [37]318 (n = 158 intervention group, n = 160 control—cow’s milk)VD fortified milk (70 IU/100 mL)Children aged 12–36 months for 20 weeksDecreased VD deficiency rate in the intervention group and increased rate in the control group (−11.8% vs. +11.4%); mean 25(OH)D level in the intervention group >30 ng/mLNo differences in the number or severity between the two groups

### 3.2. Mother Supplementation

Seven studies focused on the exclusive maternal supplementation of VD and included a total of 652 mothers (Table 2; [38,39,40,41,42,43,44]). In four trials, mothers were provided with a daily dose of VD [39,40,43,44], while three studies evaluated the efficacy of a monthly VD administration [38,41,42]. One study compared a single high dose administration given to mothers at delivery to an equivalent dose split over 28 days and found no differences in VD sufficiency neither in mothers nor infants at one month of life [44].

Two studies compared a cohort of lactating mothers receiving a high dose of VD according to two different schedules (6400 IU/d and 120,000 IU monthly in the two studies) with two cohorts of children receiving a dose of 400 IU/d [40,41]. Both studies found no significant differences in infants’ VD serum levels between intervention and control groups of children receiving 400 IU/d. Maternal supplementation of a 2000 IU or 2400 IU daily dose was less efficient than 400 IU daily treatment given to the child in preventing VD insufficiency in the offspring, even though it was capable of producing appropriate VD levels in the mothers’ blood [39,40]. These findings show that the dosages administered ranged from 400 IU/d to 150,000 IU given once, with no safety issues in neither mothers nor infants. A high dosage of 6400 IU daily or >100,000 IU monthly supplementation given to lactating mothers was shown to be equally effective as daily VD supplementation of 400 IU/d given the child to achieve VD serum levels >20 ng/mL for the first 4–6 months of life.
healthcare-10-01023-t002_Table 2Table 2Studies investigating Vitamin D supplementation for mothers.References (Authors, Year, Country)Patients, nInterventions (VD Prophylaxis)Timing of the Study (Since Delivery)Main OutcomeAdverse EventsTrivedi et al. 2020, India [38]114 (n = 58 intervention group, n = 56 placebo group)60,000 IU postpartum and at 6, 10, and 14 weeks (240,000 IU in total)From 0 to 6 months48.1% reduction in the risk of VD insufficiency (<20 ng/mL) at 6 months of age in the intervention groupNoneHollis et al. 2004, USA [39]18 (n = 9 group 1, n = 9 group 2)2000 or 4000 IU/dFrom 1 to 4 monthsSufficient VD to ensure adequate nutritional VD status for both mothers and nursing infants in group 2; adequate levels were not reached in group 1NoneHollis et al. 2015, USA [40]123 mother–infant pairs (n = 47 group 1, n = 28 group 2, n = 48 group 3)Mothers received 400, 2400, or 6400 IU/d (groups 1, 2 and 3, respectively); infants of the 400 IU group received 400 IU/d; infants in the 2400 and 6400 IU groups received placeboFrom 1 to 6 monthsNo difference between group 3 and group 1 in terms of efficacy; group 2 was excluded from final analysisNo difference among the groupsChandy et al. 2016, India [41]51 mothers, 47 infants, 54 pairs 120,000 IU/month oral (mothers, group 1);400 IU/d (infants, group 2); placebo for both mothers and infants (pairs, group 3)From 0 to 9 monthsNo difference in raising infant serum 25(OH)D in the first 9 months of life between groups 1 and 2; groups 1 and 2 had significantly higher VD sufficiency rates when compared with group 3NoneWheeler et al. 2016, New Zeeland [42]90 (n = 30 group 1, n = 30 group2, n = 30 group 3)50,000 IU (group 1), 100,000 IU (group 2), placebo monthly (group 3)From week 4 to week 20 postpartumHigher prevalence of VD insufficiency in group 3 than groups 1 and 2 (26% vs. 4% vs. 0, respectively)NoneNaik et al. 2017, India [43]115 (n = 56 intervention group, n = 59 placebo group)600,000 IU over 10 days in a dose of 60,000 IU/dayFrom 0 to 6 monthsDoubled serum VD levels in children of supplemented mothers (with levels >20 ng/mL) at 6 months of ageNoneOberhelman et al. 2013, USA [44]40 (n = 20 group 1, n = 20 group 2) 5000 IU/d for 28 days (group 1) or 150,000 IU once monthly (group 2)From day 0 to day 28No difference in VD breast milk concentrations or VD sufficiency at 1 month between the groupsNone

### 3.3. Combined Mother and Child Supplementation

Three studies evaluated the efficacy of combined supplementation for mother and child [45,46,47]. Table 3 describes these studies, which included a total of 260 mother–infant pairs. In one study, 600 IU/d VD maternal plus 400 IU/d child supplementation was compared to 6000 IU/d supplementation given just to mothers [45]. There were no significant differences between the two groups, which had similar rates of VD sufficiency. Another study comparing 6400 IU mother daily supplementation to 400 IU mother plus 300 IU infant supplementation found no differences in infant VD serum levels, with both achieving an average infant VD concentration of >40 ng/mL [47]. Another study comparing two groups of infants receiving 400 IU/d of VD for 6 months and moms receiving 1200 or 400 IU daily revealed no differences in VD sufficiency or insufficiency [46].

## 4. Discussion

This systematic review discusses the most recent evidence in the literature about the efficacy of different methods of VD supplementation on circulating VD levels to limit deficiency in infants and toddlers. No significant differences emerged in terms of safety and efficacy among proper VD supplementation regimens given directly to the mother, the child or both.

The prevalence of VD deficiency observed among breastfed infants throughout the world is higher in those who lack an adequate sunlight exposure due to limited time spent outdoors or who do not follow regular VD supplementation regimens [48,49,50]. For all of these reasons, VD deficiency in infants and toddlers can be considered a frequent problem even in healthy subjects, with a prevalence ranging from 12% to 40% [51]. VD deficiency could represent a potential problem in infants exclusively breastfed by mothers having VD intakes of 400 IU/day, the amount normally assumed during pregnancy [40], and who do not receive direct and proper supplementation, as shown by a Norwegian study, which found a prevalence of 31% of VD deficiency in a population of 52 infants at 3 months of life, compared with a prevalence of 13% in their mothers at the same time [52]. VD deficiency can impact the quality of life of the child and particularly of the infant in the short and long term, and VD supplementation has been identified as a necessary and effective instrument for preventing VD deficiency. Possible consequences of VD deficiency include frequent infections, a higher risk of bone fractures and poor growth [53,54,55]. Indeed, the implementation of VD supplementation as a public health policy has been shown to reduce the prevalence of rickets in early childhood worldwide [56].

The choice of including subjects aging 12 to 36 months, rather than only infants, was made since they have different nutritional intakes, dietary habits and lifestyles. Indeed, just a few studies evaluated VD needs during the first years of life, and little is known on how the VD status changes after breastfeeding is discontinued. Additionally, the European Food Safety Authority set an adequate intake of 15 μg/d for children aged 1–17 years, not excluding the 12–24 months range of age and based on data mostly collected on adults [7,57]. Furthermore, because fortified milks are only recommended after the age of 12 months [58], and because a large number of factors affect the response to supplementation, such as baseline status at the end of breastfeeding, ethnicity compliance, latitude and diet, more studies on toddlers are needed to understand an eventually widespread VD deficiency at this age.

Several studies have been conducted on a possible association between the supplementation of VD and the reduced risk of developing autoimmune diseases, such as type 1 diabetes [59,60,61] and inflammatory bowel diseases [62,63,64,65], with conflicting results. Experimental data suggest that VD is able to modulate monocyte, macrophage and dendritic-cell response, and production of interleukins [66]. Recent studies suggest an involvement of VD in the suppression of T-lymphocyte proliferation and adaptive immune system, causing a shift from a Th1 to a Th2 phenotype and a subsequent alteration in T-cell differentiation and maturation, inducing T regulatory cells function and immune self-tolerance [67,68]. B lymphocytes also express VD receptors, which, when activated, can inhibit the differentiation into plasma cells and modulate immunoglobulin production [69]. All of these effects could explain the possible connection between variable VD serum levels and the probability of developing an autoimmune disease [70].

The present review suggests that exclusive high-dose supplementation provided to mothers and a proper administration given to the child can both be deemed suitable approaches for the prevention of VD deficiency in exclusively breastfed infants. Accordingly, the significant maternal VD supplementation of 6400 IU/day or >100,000 IU on a monthly schedule has been found to be able to safely supply adequate amounts of VD to breast milk [40,44], completely satisfying infant requirements and offering an alternative strategy to direct infant daily supplementation, which may be limited in efficacy due to caregivers’ low compliance [35,48,71]. The same conclusions have been obtained by milk fortification or a single 50,000 IU dose given to the child. In the collected studies, high doses given daily or monthly to either mothers or children had no dangerous effects in terms of safety or adverse outcomes, with similar efficacy outcomes [34,38,39,41,42,43,44,47]. Regarding assuming excessive or even toxic accumulation in the body, no evidence for prophylactic dosages is given, probably due to the lipophilic nature of VD with slow processes of turnover in fat deposits [72].

According to most of the international recommendations and global consensus about the prevention of rickets, all infants should be orally supplemented with 400 IU/day of VD from birth to 12 months of age, regardless of feeding and nutrition types [73]. The efficacy of trials conducted by daily administration may still suffer from the previously described bias of poor compliance in real-world experiences, resulting in lower effectiveness [74].

Considering that the compliance with daily VD supplementation in the first years of life is variable in different countries, and that rates of withdrawal and adherence change during the first months of life, an increased awareness about the correct posology and doses of VD administration is needed [22,23,24]. The main factors affecting VD levels in infants are related to the mother’s serum concentrations during pregnancy and lactation, the length of sun exposure of the child and the total intake assumed by diet or supplementation [75,76,77]. While fish, lipids and dairy products are the main sources of VD for humans, breast milk is a limited source of this micronutrient [1,78]. VD concentrations in breast milk are commonly expressed as antirachitic activity, which is calculated from the measurement of milk VD and 25(OH)D concentrations and then translated to biological activity using reference data [79]. Furthermore, the principal form of VD transferred from maternal circulation to breast milk is represented by cholecalciferol, with very few amounts of 25(OH)D passing directly via breast milk [80]. Another limiting step of VD passage in any form is the rapid conversion of cholecalciferol to 25(OH)D in mothers’ liver, which may be one of the main reasons for the low ratio between maternal serum 25(OH)D and breast milk VD concentration, which is approximately 0.2 [81]. Accordingly, mothers with low 25(OH)D serum concentrations supply even less VD to their infants, whereas higher VD levels in breast milk positively correlate with high serum levels of VD in mothers [2,3,4]. Therefore, if VD is not supplemented directly to the child, the mother’s VD intake during lactation should be higher than during pregnancy, when the ratio between serum and cord blood levels is in the range 0.5–0.8 [82,83].

According to the European Society for Pediatric Gastroenterology Hepatology and Nutrition (ESPGHAN) guidelines, all infants during the first year of life should receive 400 IU/day oral supplementation of VD, resulting in a minimal serum concentration of 20 ng/mL [84]. Beyond this age, seasonal variations in sunlight should be considered within a national policy for supplementation or fortification. Nevertheless, there is evidence that infants who receive a mixture of human milk and formula should also receive a VD supplement of 400 IU/day to ensure adequate intakes and support circulating VD [85]. As infants and toddlers are weaned from breastfeeding and/or formula, fortified milk and formulae or continuous VD supplementation should be encouraged to provide at least 400 IU/day of VD [58]. As an example, it has been shown that formula-fed infants receiving more than 1 L of VD-enriched formula do not need a higher VD intake through supplements [85,86].

To promote compliance to VD supplements, the fortification of commonly used food products has also been suggested [36,37,87]. Indeed, VD fortified products may be more efficient than intermittent supplements in maintaining adequate stores [37]. Enriched milk could represent an effective method of delivering VD because of its wide availability and acceptance. Accordingly, a prospective, randomized, double-blind study conducted in Germany on 92 children between 2 and 6 years of age has shown that consumption of milk fortified with VD can be a simple and safe nutritional measure to prevent decreasing serum 25(OH)D concentrations in periods or places with poor sunlight [88].

One of the strength of this review is the specific focus on the efficacy outcomes of the studies included. We evaluated the VD status of children not in terms of higher serum levels achieved by different cohorts, but in terms of real VD deficiency prevalence in the first years of life. Moreover, this review adds the evaluation of the safety outcomes. Additionally, the exclusion of trials that included high prenatal supplementation and/or VD deficient mothers at baseline reduced the possible risk of bias. Another strength of this study may be considered to be the inclusion of both breastfed infants and toddlers, as well as the rarely described weekly/monthly administration schedules.

An important limitation of this systematic review is the impossibility to perform a formal meta-analysis using a quantitative comparison, due to the wide heterogeneity of study designs, with different dosing frequencies and amounts, different baseline VD status at randomization, inclusion criteria and follow-up length.

## 5. Conclusions

In conclusion, different supplementation schedules given to mothers in RCTs showed similar efficacy and safety, and no adverse effects have been observed in children receiving high VD dosages, despite the fact that many of these strategies have not yet been implemented in clinical practice. VD supplementation should be revised according to the different settings, offering multiple options to families, or evaluating the best choice case-by-case, especially for subjects who are more prone to poor adherence to daily supplementation. Moreover, considering the wide variability in the causes that could lead to this poor adherence in culturally and economically diverse settings and countries, specific strategies that could help to increase adherence rates are not simple to be identified, and case-by-case reasoning should be performed.

Additional dose–response studies on other types of supplementation strategies are then needed, with a specific focus on the 12–36 months range, given that dosages exceeding 400 IU/d did not provide additional benefits on the health and bone status in the first year of life. International guidelines should reconsider supplementation and milk fortification procedures in view of real-world compliance, which seems to be lower than expected, particularly beyond the first months of life.

## Figures and Tables

**Figure 1 healthcare-10-01023-f001:**
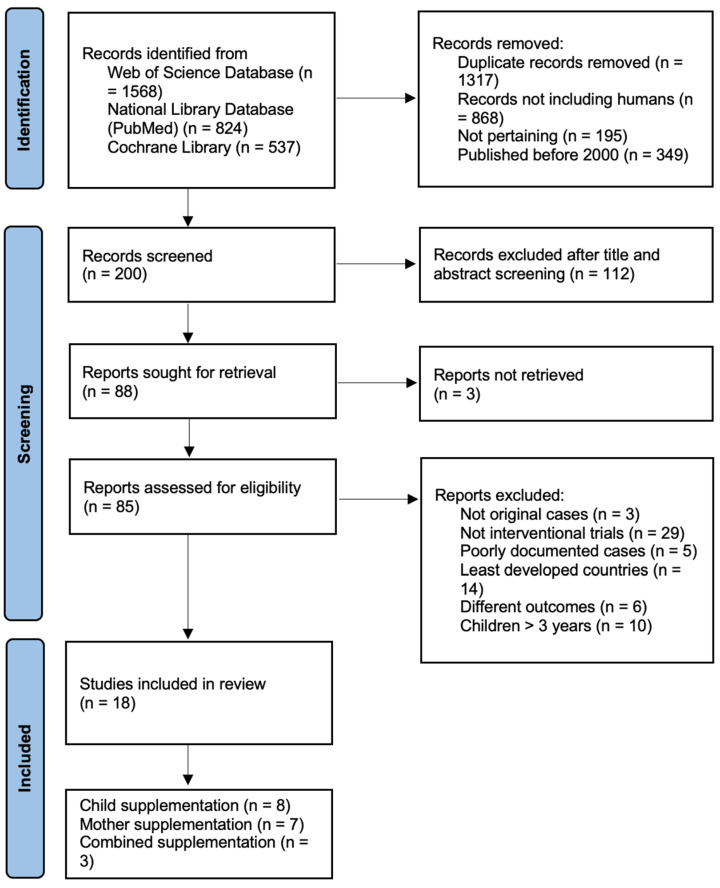
PRISMA flow diagram for systematic reviews.

**Table 3 healthcare-10-01023-t003:** Studies investigating Vitamin D supplementation for both mother and child.

References (Authors, Year, Country)	Patients, n	Intervention (VD Prophylaxis)	Timing of the Study (Since Delivery)	Main Outcome	Adverse Events
Dawodu et al. 2019, Qatar [45]	208 (n = 56 pairs group 1, n = 48 pairs group 2)	Maternal supplementation of 6000 IU/d (group 1) or maternal supplementation of 600 IU/d plus infant supplementation of 400 IU/d (group 2)	From 0 to 6 months	Slightly higher serum 25(OH)D levels in infants of mother from group 1; similar rates of VD sufficiency (>20 ng/mL) between the two groups	None
Czech-Kowalska et al. 2014, Poland [46]	274 (n = 67 pairs group 1, n = 70 group 2)	Maternal supplementation of 1200 IU/d (group 2) or 400 IU/d (group 1); 400 IU/d for all infants	From 0 to 6 months	Comparable prevalence of infants’ VD deficiency and sufficiency	Not available
Wagner et al. 2006, USA [47]	38 (n = 9 pairs in the mother-alone supplementation group, n = 10 pairs in both mother and child supplementation group)	Maternal supplementation of 400/d or 6400 IU/d; 300 IU/d supplementation for infants in 400 IU group, placebo for infants in 6400 IU group	From 1 to 7 months of child’s life	Equivalent infant levels of VD in both groups at 7 months of life.	None

## Data Availability

Not applicable.

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
