# Peer review of "Different Vitamin D Supplementation Strategies in the First Years of Life: A Systematic Review"

_healthcare, 2022, doi:10.3390/healthcare10061023_

Round 1
Reviewer 1 Report
Overall, I enjoyed reviewing this manuscript. It was well organized. Below are my comments/edits:
Lines 77-79: remove parenthesis and change CAPS to lower case
Line 92: it would be helpful to add a sentence or two on why the two investigators were chosen. What were the qualifications? Did they have over 100 publications? Expertiese in human health for over 25 years? etc
Line 101: what were parameters for a high evaluation?
Lines 115 - 119: I see the intention was for a total of 18 articles; however, only 16 are accounted for by countries.
References: Should this be alphabetized in order?
Author Response
Revision paper ID 1693592 entitled "Different Vitamin D Supplementation Strategies in the First Years of Life: A Review”.
Dear Editorial Board of Healthcare, we thank the Reviewers for all the important comments. We have revised the manuscript in accordance with the indications. Changes have been performed throughout the whole text. Specifically following changes have been made and the point-by-point explanation to the reviewer comments may be given.
Reviewers’ Comments:
Reviewer 1: Overall, I enjoyed reviewing this manuscript. It was well organized. Below are my comments/edits:
Lines 77-79: remove parenthesis and change CAPS to lower case
Line 92: it would be helpful to add a sentence or two on why the two investigators were chosen. What were the qualifications? Did they have over 100 publications? Expertise in human health for over 25 years? etc
Line 101: what were parameters for a high evaluation?
Lines 115 - 119: I see the intention was for a total of 18 articles; however, only 16 are accounted for by countries.
References: Should this be alphabetized in order?
- We thank the Reviewer for his suggestions and appreciation. We have modified the manuscript accordingly to his precise comments, when possible. We left the search string as it has been originally used during the search process.
Reviewer 2: As children needs vitamin d supplementation as recommended by (ESPGHAN) guidelines, maternal vitamin d supplementation seems like a necessity. Well-designed study.
As more and more people are vitamin d deficient due to lack of sun exposure, vitamin d supplementation is a must for all ages. Vitamin d fortified milk seems to be an effective strategy. Since vitamin d engages in very early stages of embryo development, maternal vitamin d supplementation is also very important.
- We thank the Reviewers for this constructive feedback and compliment. We have specifically described possible negative consequences of VD deficiency and its importance in the supplementation of the infant.
Reviewer 3: The title is clear and in correlation with the subject, the methods well-described and the conclusion is right and in correlation with the discussion. The question and the "take-home message” are that the physician may propose different strategies to assure sufficiently vitamin D in babies and children till 24 months.
- We agree with the reviewer about the conclusion of this study and its “take-home message”. We would like to express our thanks to the reviewer for his appreciation to our study and his compliments.
Reviewer 4:
Comment # 1: I would like to congratulate the authors on their work, as it explores from a metareview approach the diverse strategies of vitamin D supplementation on infants, attempting to discern possible discrepancies on current criteria. The manuscript is well-drafted, scientifically rigorous and sound, as well as clear and concise. Nonetheless, there are some points that would need clarification. Comments are provided below:
The Introduction section provides a meaningful background on the topic and current advances, discussing discrepancies and accepted dosages.
- We sincerely thank the reviewer for his words and constructive feedbacks. We modified the text according to all the comments written below.
Minor comments:
L40:Could the authors elaborate a bit on these health outcomes, especially regarding early development?
L43: What do the authors mean by “scientific societies”?
L47-48: The statement is a bit unclear. Could the authors rephrase it?
L66-67: I don’t think this phrase is very clear. Could the authors rephrase a bit or perhaps elaborate a bit on this?
The Methods appear clear and easily reproducible, with almost no gaps. I have a question however, regarding L91-94 about the “two authors” assessing the quality and acceptance of the studies retrieved, and this “third reviewer”. It is unclear how this allows for a more accurate analysis.
The Results are excellently clear and presented in a very ordered manner. The use of tables presenting specific details on each study and type also aids in the interpretation of the results and the general findings retrieved. Likewise, the text is very concise on detailing relevant aspects related to each treatment option and key findings of each study, especially regarding dosage and vitamin D levels quantified.
Albeit the Discussion section is generally well structured, I find some conflicting points to it.
General recommendations on vit. D intakes appear the same (L242-244; 268-271) but no correlation is stablished between them, although the first paragraph cites “international recommendations” and the second one a specific organism.
L277-278: Do formulas have a very specific concentration of vitamin D? It should perhaps be best to specify vitamin D intake instead of volume of formula.
I don’t quite understand why reference 90 wasn’t included in the meta-analysis, since it seems it accomplishes all eligible criteria. Its results seem meaningful too.
I find a statement discussing the prevalence of vit. D deficiencies lacking in the discussion. Could the authors briefly comment on strategies to improve treatment adherence?
Altogether, the manuscript is of interest and high quality. I would recommend its acceptance after these few issues are resolved. Therefore, I suggest minor revisions.
- We sincerely thank the reviewer for his great interest in our study, with a deep analysis and great suggestions. We have then highlighted the characteristics of the study, as suggested, further describing them in our text. All the sentences that were unclear according to the reviewer were re-written, in order to result more consistent. Even if reference 90 indicated a meaningful study, it just included children >2 years of age, and it did not meet our age criteria (point iii). Conclusion has been integrated with comments about the difficulty to identify strategies for increase the adherence rates.
Reviewer 5:
Comment # 1: The authors describe the interesting subject that we need new revision about supplementation of vitamin D3 because they presented the systematic review where there are divergences in the dose level of vitamin D3 and its effects. The manuscript is thoughtful and well written.
I recommend publishing the article with minor corrections.
My comments on the manuscript are described below:
- Your Title of manuscript Different vitamin D supplementation strategies in the first years of life: a review”.
Should there be "the systematic review" in the title of the manuscript instead of "review"
- please specify precisely the date range covered by the systematic review, not only years.
- We sincerely thank the reviewer for his appreciation and constructive feedbacks. We modified the text according to all the comments reported, modifying the tile and better specifying the date range.
As you notice, we agreed with all the comments raised by the reviewers, changing the title and replying to all their feedbacks. We would like to take this opportunity to express our sincere thanks to the reviewers who identified areas of our manuscript that needed corrections or modification.
Reviewer 2 Report
As children needs vitamin d supplementation as recommended by (ESPGHAN) guidelines, maternal vitamin d supplementation seems like a necessity.
Well designed study.
As more and more people are vitamin d deficient due to lack of sun exposure, vitamin d supplementation is a must for all ages.
Vitamin d fortified milk seems to be an effective strategy.
Since vitamin d engages in very early stages of embryo developement, maternal vitamin d supplementation is also very important.
Author Response

(The authors gave the same response as above.)

Reviewer 3 Report
The tilte is clear and in correlation with the subject, the methods well-described and the conclusion is right and in correlation with the discussion. The question and the " take-home message " is that the physician may propose different strategies to assure sufficiently vitamin D in babies and children till 24 months.
Author Response

(The authors gave the same response as above.)

Reviewer 4 Report
I would like to congratulate the authors on their work, as it explores from a metareview approach the diverse strategies of vitamin D supplementation on infants, attempting to discern possible discrepancies on current criteria. The manuscript is well-drafted, scientifically rigorous and sound, as well as clear and concise. Nonetheless, there are some points that would need clarification. Comments are provided below:
The Introduction section provides a meaningful background on the topic and current advances, discussing discrepancies and accepted dosages. I have however some minor comments.
L40:Could the authors elaborate a bit on these health outcomes, especially regarding early development?
L43: What do the authors mean by “scientific societies”?
L47-48: The statement is a bit unclear. Could the authors rephrase it?
L66-67: I don’t think this phrase is very clear. Could the authors rephrase a bit or perhaps elaborate a bit on this?
The Methods appear clear and easily reproducible, with almost no gaps. I have a question however, regarding L91-94 about the “two authors” assessing the quality and acceptance of the studies retrieved, and this “third reviewer”. It is unclear how this allows for a more accurate analysis.
The Results are excellently clear and presented in a very ordered manner. The use of tables presenting specific details on each study and type also aids in the interpretation of the results and the general findings retrieved. Likewise, the text is very concise on detailing relevant aspects related to each treatment option and key findings of each study, especially regarding dosage and vitamin D levels quantified.
Albeit the Discussion section is generally well structured, I find some conflicting points to it.
General recommendations on vit. D intake appear the same (L242-244; 268-271) but no correlation is stablished between them, although the first paragraph cites “international recommendations” and the second one a specific organism.
L277-278: Do formulas have a very specific concentration of vitamin D? It should perhaps be best to specify vitamin D intake instead of volume of formula.
I don’t quite understand why reference 90 wasn’t included in the meta-analysis, since it seems it accomplishes all eligible criteria. Its results seem meaningful too.
I find a statement discussing the prevalence of vit. D deficiencies lacking in the discussion. Could the authors briefly comment on strategies to improve treatment adherence?
Altogether, the manuscript is of interest and high quality. I would recommend its acceptance after these few issues are resolved. Therefore, I suggest minor revisions.
Author Response

(The authors gave the same response as above.)

Reviewer 5 Report
Thank you that you give me the opportunity to review this manuscript “
“Different vitamin D supplementation strategies in the first years of life: a review”.
The authors describe the interesting subject that we need new revision about supplementation of vitamin D3 because they presented the systematic review where there are divergences in the dose level of vitamin D3 and its effects. The manuscript is thoughtful and well written.
I recommend publishing the article with minor corrections.
My comments on the manuscript are described below:
1. Your Title of manuscript Different vitamin D supplementation strategies in the first years of life: a review”.
Should there be "the systematic review" in the title of the manuscript instead of "review"?
2. please specify precisely the date range covered by the systematic review, not only years.
Author Response
Revision paper ID 1693592 entitled "Different Vitamin D Supplementation Strategies in the First Years of Life: A Review”.
Dear Editorial Board of Healthcare, we thank the Reviewers for all the important comments. We have revised the manuscript in accordance with the indications. Changes have been performed throughout the whole text. Specifically following changes have been made and the point-by-point explanation to the reviewer comments may be given.
Reviewers’ Comments:
Reviewer 1: Overall, I enjoyed reviewing this manuscript. It was well organized. Below are my comments/edits:
Lines 77-79: remove parenthesis and change CAPS to lower case
Line 92: it would be helpful to add a sentence or two on why the two investigators were chosen. What were the qualifications? Did they have over 100 publications? Expertise in human health for over 25 years? etc
Line 101: what were parameters for a high evaluation?
Lines 115 - 119: I see the intention was for a total of 18 articles; however, only 16 are accounted for by countries.
References: Should this be alphabetized in order?
- We thank the Reviewer for his suggestions and appreciation. We have modified the manuscript accordingly to his precise comments, when possible. We left the search string as it has been originally used during the search process.
Reviewer 2: As children needs vitamin d supplementation as recommended by (ESPGHAN) guidelines, maternal vitamin d supplementation seems like a necessity. Well-designed study.
As more and more people are vitamin d deficient due to lack of sun exposure, vitamin d supplementation is a must for all ages. Vitamin d fortified milk seems to be an effective strategy. Since vitamin d engages in very early stages of embryo development, maternal vitamin d supplementation is also very important.
- We thank the Reviewers for this constructive feedback and compliment. We have specifically described possible negative consequences of VD deficiency and its importance in the supplementation of the infant.
Reviewer 3: The title is clear and in correlation with the subject, the methods well-described and the conclusion is right and in correlation with the discussion. The question and the "take-home message” are that the physician may propose different strategies to assure sufficiently vitamin D in babies and children till 24 months.
- We agree with the reviewer about the conclusion of this study and its “take-home message”. We would like to express our thanks to the reviewer for his appreciation to our study and his compliments.
Reviewer 4:
Comment # 1: I would like to congratulate the authors on their work, as it explores from a metareview approach the diverse strategies of vitamin D supplementation on infants, attempting to discern possible discrepancies on current criteria. The manuscript is well-drafted, scientifically rigorous and sound, as well as clear and concise. Nonetheless, there are some points that would need clarification. Comments are provided below:
The Introduction section provides a meaningful background on the topic and current advances, discussing discrepancies and accepted dosages.
- We sincerely thank the reviewer for his words and constructive feedbacks. We modified the text according to all the comments written below.
Minor comments:
L40:Could the authors elaborate a bit on these health outcomes, especially regarding early development?
L43: What do the authors mean by “scientific societies”?
L47-48: The statement is a bit unclear. Could the authors rephrase it?
L66-67: I don’t think this phrase is very clear. Could the authors rephrase a bit or perhaps elaborate a bit on this?
The Methods appear clear and easily reproducible, with almost no gaps. I have a question however, regarding L91-94 about the “two authors” assessing the quality and acceptance of the studies retrieved, and this “third reviewer”. It is unclear how this allows for a more accurate analysis.
The Results are excellently clear and presented in a very ordered manner. The use of tables presenting specific details on each study and type also aids in the interpretation of the results and the general findings retrieved. Likewise, the text is very concise on detailing relevant aspects related to each treatment option and key findings of each study, especially regarding dosage and vitamin D levels quantified.
Albeit the Discussion section is generally well structured, I find some conflicting points to it.
General recommendations on vit. D intakes appear the same (L242-244; 268-271) but no correlation is stablished between them, although the first paragraph cites “international recommendations” and the second one a specific organism.
L277-278: Do formulas have a very specific concentration of vitamin D? It should perhaps be best to specify vitamin D intake instead of volume of formula.
I don’t quite understand why reference 90 wasn’t included in the meta-analysis, since it seems it accomplishes all eligible criteria. Its results seem meaningful too.
I find a statement discussing the prevalence of vit. D deficiencies lacking in the discussion. Could the authors briefly comment on strategies to improve treatment adherence?
Altogether, the manuscript is of interest and high quality. I would recommend its acceptance after these few issues are resolved. Therefore, I suggest minor revisions.
- We sincerely thank the reviewer for his great interest in our study, with a deep analysis and great suggestions. We have then highlighted the characteristics of the study, as suggested, further describing them in our text. All the sentences that were unclear according to the reviewer were re-written, in order to result more consistent. Even if reference 90 indicated a meaningful study, it just included children >2 years of age, and it did not meet our age criteria (point iii). Conclusion has been integrated with comments about the difficulty to identify strategies for increase the adherence rates.
Reviewer 5:
Comment # 1: The authors describe the interesting subject that we need new revision about supplementation of vitamin D3 because they presented the systematic review where there are divergences in the dose level of vitamin D3 and its effects. The manuscript is thoughtful and well written.
I recommend publishing the article with minor corrections.
My comments on the manuscript are described below:
- Your Title of manuscript Different vitamin D supplementation strategies in the first years of life: a review”.
Should there be "the systematic review" in the title of the manuscript instead of "review"
- please specify precisely the date range covered by the systematic review, not only years.
- We sincerely thank the reviewer for his appreciation and constructive feedbacks. We modified the text according to all the comments reported, modifying the tile and better specifying the date range.
As you notice, we agreed with all the comments raised by the reviewers, changing the title and replying to all their feedbacks. We would like to take this opportunity to express our sincere thanks to the reviewers who identified areas of our manuscript that needed corrections or modification.
This manuscript is a resubmission of an earlier submission. The following is a list of the peer review reports and author responses from that submission.
Round 1
Reviewer 1 Report
This is a well-written systematic review on vitamin D supplementation trials for infants and children up to 36 months. My main concern with this paper is that there are a number of RCTs conducted in the past 10 years which are missing including Gallo et al. 2013, Ziegler et al. 2014, and Cooper et al. 2016 (MAVIDOS). The Tan et al 2020 Cochrane Review included these trials, as well as others, and it is unclear why there were excluded as they seem to fit the eligibility criteria. Further, it is not clear what this systematic review adds to the current literature considering the other systematic reviews and meta-analyses conducted recently on this topic. The authors need to include a stronger justification for their study in the introduction, and also address why a meta-analysis was not included. In addition, justification for including infants/ children <36 months is needed infants <12 months will have different nutritional intakes i.e. vitamin D fortified cow’s milk is not recommended until 1 year of age. There are a number of factors that affect response to supplementation and the authors have not addressed this in their results. This includes baseline status of infant/mothers but, also other factors such as race/ethnicity compliance, latitude, etc.
Author Response
Comment # 1: This is a well-written systematic review on vitamin D supplementation trials for infants and children up to 36 months. My main concern with this paper is that there are a number of RCTs conducted in the past 10 years which are missing including Gallo et al. 2013, Ziegler et al. 2014, and Cooper et al. 2016 (MAVIDOS). The Tan et al 2020 Cochrane Review included these trials, as well as others, and it is unclear why they were excluded as they seem to fit the eligibility criteria.
- We have included the proposed papers that met the inclusion and exclusion criteria (see also the reply to Comment #4, Reviewer 2). Some other papers, such as that from Cooper et al., have been excluded since they did not fulfill the inclusion criteria (i.e., prenatal supplementation).
Comment # 2: Further, it is not clear what this systematic review adds to the current literature considering the other systematic reviews and meta-analyses conducted recently on this topic.
- We thank the Reviewer for this suggestion. The strengths of the present review have been highlighted in the discussion section.
Comment # 3: The authors need to include a stronger justification for their study in the introduction, and also address why a meta-analysis was not included.
- We have better clarified the scientific background from which the idea of conducting this study originated. The heterogeneity of dosages, populations, and schedules, as well as the different follow-up times, precluded us from conducting a formal and quantitative analysis. We have reported this aspect among the limitations.
Comment # 4: In addition, justification for including infants/ children <36 months is needed. infants <12 months will have different nutritional intakes i.e. vitamin D fortified cow’s milk is not recommended until 1 year of age.
- We have provided justification for this topic, better describing the choice of the range of age.
Comment # 5: There are a number of factors that affect response to supplementation and the authors have not addressed this in their results. This includes baseline status of infants/mothers but also other factors such as race/ethnicity compliance, latitude, etc.
- We have included and described the different factors that may alter VD values and absorption.
As you notice, we agreed with all the comments raised by the reviewers. We would like to take this opportunity to express our sincere thanks to the reviewers who identified areas of our manuscript that needed corrections or modification.
Reviewer 2 Report
The article “Vitamin D supplementation strategies in the first years of life: a systematic review” has focused on relevant health problem – the need of vitamin D supplementation during the first years of life. It is well known, that vitamin D characterised by pleiotropic action, acts positive to human organism, thus its adequate level should be maintained. Unfortunately, the decreased vitamin D serum concentration among small children is very common.
Issues that can be improved:
- The article is interesting, although in introduction or discussion section, besides vitamin D supplementation, recommendation concerning diet and vitamin D production under solar radiation could be written. Also the different conditions such as genetic, environmental and personal factors influencing vitamin D concentration could be discussed.
- Abstract could be more specific – more results should be given.
- In introduction also more specific information regarding to vitamin D functions and the possible negative impact of its deficiency should be described (line 37 to 43).
- In discussion section (line 174 to 180) more information about negative consequences correlated with vitamin D deficiency in children should be given.
- No strengths of the study was given.
- English and scientific language could be improved.
- References should be prepared according to Instructions for Authors of the Journal.
Author Response
Comment # 1: The article is interesting, although in the introduction or discussion section, besides vitamin D supplementation, recommendations concerning diet and vitamin D production under solar radiation could be written. Also, the different conditions such as genetic, environmental and personal factors influencing vitamin D concentration could be discussed.
- To address this concern, we have widely reworked the introduction section, including the different conditions that may influence VD levels in the first years of life.
Comment # 2: Abstract could be more specific – more results should be given.
- We appreciate the Reviewer’s critical feedback. The abstract has been rewritten to include more detailed results.
Comment # 3: In introduction also more specific information regarding vitamin D functions and the possible negative impact of its deficiency should be described (line 37 to 43).
- We have included more information on the risks and possible negative effects of VD deficiency, as suggested (see discussion section).
Comment # 4: In the discussion section (line 174 to 180) more information about negative consequences correlated with vitamin D deficiency in children should be given.
- We thank the Reviewers for this constructive feedback. We have specifically described possible negative consequences of VD deficiency.
Comment # 5: No strengths of the study were given.
- We have highlighted the strengths of the study, as suggested, further describing the aim and conclusions of our paper.
Comment # 6: English and scientific language could be improved.
- English has been checked throughout the text by a native English speaker.
Comment # 7: References should be prepared according to Instructions for Authors of the Journal.
- We have edited references according to the Journal instructions. More trials that we missed in our search have been added to our review, and the discussion has been implemented and corrected in order to be clearer. The Methods section has been re-written, with a specific paragraph on inclusion and exclusion criteria that we used.
Reviewer 3 Report
This is a potential interesting investigation, however, there are some major issues in the study:
- The aim of the study is not clearly defined and vaguely stated. What are the primary and secondary endpoints of the study?
- The study should have been registered before conducting it on PROSPERO. Although, this is too late to correct it should be considered in the overall assessment of the quality of the study
- What are the inclusion and exclusion criteria of the studies? This is only shown in the exclusion process of the flowchart, however, not mentioned anywhere in the methods section. This is very important for a systematic review to clearly define to the readers before reading into it.
- I question the search process. To my knowledge there are several other studies potential for inclusion with vitamin D intervention among children and pregnant women. Why are these not included? I would expect a lot more than those excluded from the final stage of the flowchart
- The flowchart should be divided into maternal and child studies
- It is difficult to conclude anything based on these few heterogenous studies included in this trial. I don't agree with the conclusions stated except from the safety endpoint which seems consistent throughout the studies.
- Why is no meta-analysis performed? This should be done
Author Response
Comment # 1: This is a potential interesting investigation, however, there are some major issues in the study:
The aim of the study is not clearly defined and vaguely stated. What are the primary and secondary endpoints of the study?
- We have better defined the aim of the study, as suggested.
Comment # 2: The study should have been registered before conducting it on PROSPERO. Although, this is too late to correct it should be considered in the overall assessment of the quality of the study
- We thank the Reviewer for this suggestion. We have not implemented PROSPERO registration is an important suggestion, but not mandatory according to the PRISMA guidelines. Currently, many months are necessary before having the approval of a protocol, and for this reason we did not record it. However, in case the editor feels that a registration is necessary, we are available to register the protocol on a different database.
Comment # 3: What are the inclusion and exclusion criteria of the studies? This is only shown in the exclusion process of the flowchart, however, not mentioned anywhere in the methods section. This is very important for a systematic review to clearly define to the readers before reading into it.
- Inclusion and exclusion criteria have been specified in the Methods, as suggested.
Comment # 4: I question the search process. To my knowledge there are several other studies potential for inclusion with vitamin D intervention among children and pregnant women. Why are these not included? I would expect a lot more than those excluded from the final stage of the flowchart
- We have checked our literature research once again and found out some more clinical studies. The search flowchart has been updated accordingly.
Comment # 5: The flowchart should be divided into maternal and child studies
- We have modified the flowchart accordingly.
Comment # 6: It is difficult to conclude anything based on these few heterogenous studies included in this trial. I don't agree with the conclusions stated except from the safety endpoint which seems consistent throughout the studies.
- We have modified the conclusions in order to make them more cautious and to avoid overlooking the findings. Moreover, strengths and limitations of the study have been detailed.
Comment # 7: Why is no meta-analysis performed? This should be done
- We agree with the Reviewer that a meta-analysis would enrich the overall value of our review. However, as stated in the limitations, the heterogeneity of dosages, populations, and schedules, as well as the different follow-up times, precluded us from conducting a formal and quantitative analysis.
Round 2
Reviewer 1 Report
The authors have addressed my previous comments although, there are still significant concerns in this manuscript. The systematic review protocol is questionable as the two missing publications (Gallo et al. 2013 & Ziegler et al. 2014) were added based on my previous comment without justification for why they were omitted from the initial review. Further, the lack of meta-analysis limits the interpretation of the results. Although heterogeneity among studies would be an issue, there are statistical methods to account for this i.e. Cochran’s Q and I2 tests of heterogeneity. Different supplementation dosages, lengths of study, etc. have been combined previous meta-analyses. Finally, subgroup analysis could have been conducted to account for differences in baseline status, and possibly other factors. As there are a number of recent systematic reviews and meta-analyses conducted on this topic (i.e. Tan et al. Cochrane Database Syst Rev. 2020; O'Callaghan et al. Adv Nutr. 2020; Mimouni & Mendlovic. Curr Opin Clin Nutr Metab Care. 2021; Zittermann et al. Eur J Nutr. 2020; Cashman et al. Am J Clin Nutr. 2022), I am not sure what this current paper adds to the literature.
Reviewer 3 Report
No further comments